# JNK and p38 Inhibitors Prevent Transforming Growth Factor-β1-Induced Myofibroblast Transdifferentiation in Human Graves’ Orbital Fibroblasts

**DOI:** 10.3390/ijms22062952

**Published:** 2021-03-14

**Authors:** Tzu-Yu Hou, Shi-Bei Wu, Hui-Chuan Kau, Chieh-Chih Tsai

**Affiliations:** 1Department of Ophthalmology, Taipei Veterans General Hospital, Taipei 112, Taiwan; houtztz@gmail.com (T.-Y.H.); hckau1234@yahoo.com (H.-C.K.); 2School of Medicine, National Yang Ming University, Taipei 112, Taiwan; 3School of Medicine, National Yang Ming Chiao Tung University, Hsinchu 30010, Taiwan; 4Biomedical Commercialization Center, Taipei Medical University, Taipei 110, Taiwan; barry0110@tmu.edu.tw; 5Department of Ophthalmology, Koo Foundation Sun Yat-Sen Cancer Center, Taipei 112, Taiwan

**Keywords:** c-Jun N-terminal kinase, Graves’ orbital fibroblasts, Graves’ ophthalmopathy, mitogen-activated protein kinase, p38, transforming growth factor-β1

## Abstract

Transforming growth factor-β1 (TGF-β1)-induced myofibroblast transdifferentiation from orbital fibroblasts is known to dominate tissue remodeling and fibrosis in Graves’ ophthalmopathy (GO). However, the signaling pathways through which TGF-β1 activates Graves’ orbital fibroblasts remain unclear. This study investigated the role of the mitogen-activated protein kinase (MAPK) pathway in TGF-β1-induced myofibroblast transdifferentiation in human Graves’ orbital fibroblasts. The MAPK pathway was assessed by measuring the phosphorylation of p38, c-Jun N-terminal kinase (JNK), and extracellular-signal-regulated kinase (ERK) by Western blots. The expression of connective tissue growth factor (CTGF), α-smooth muscle actin (α-SMA), and fibronectin representing fibrogenesis was estimated. The activities of matrix metalloproteinases (MMPs) and tissue inhibitors of metalloproteinases (TIMPs) responsible for extracellular matrix (ECM) metabolism were analyzed. Specific pharmacologic kinase inhibitors were used to confirm the involvement of the MAPK pathway. After treatment with TGF-β1, the phosphorylation levels of p38 and JNK, but not ERK, were increased. CTGF, α-SMA, and fibronectin, as well as TIMP-1 and TIMP-3, were upregulated, whereas the activities of MMP-2/-9 were inhibited. The effects of TGF-β1 on the expression of these factors were eliminated by p38 and JNK inhibitors. The results suggested that TGF-β1 could induce myofibroblast transdifferentiation in human Graves’ orbital fibroblasts through the p38 and JNK pathways.

## 1. Introduction

Graves’ ophthalmopathy (GO) is an autoimmune disorder characterized by initial inflammation and later tissue expansion, remodeling, and/or fibrosis, causing cosmetically disfiguring and vision-threatening morbidities [1,2]. Orbital fibroblasts are known to be the most important effector cells [3]. The definite pathogenesis, however, is incompletely understood, especially in late-stage GO when the disease progresses into orbital soft tissue fibrosis, leading to proptosis, exposure keratopathy, diplopia, and compressive optic neuropathy, which are usually refractory to most medical treatments and require surgical rehabilitation [4,5,6].

Transforming growth factor-β (TGF-β) is believed to induce tissue remodeling and fibrosis through myofibroblast transdifferentiation in Graves’ orbital fibroblasts [6,7]. Although previous studies have confirmed the higher protein and messenger ribonucleic acid (mRNA) expression levels of TGF-β1 in Graves’ orbital tissues and orbital fibroblasts [8,9,10], the molecular mechanisms responsible for TGF-β1-induced myofibroblast transdifferentiation in GO have not been established.

TGF-β1 can promote various fibrotic diseases by stimulating either canonical (Smad-based) or noncanonical (non-Smad-based) signaling pathways. The mitogen-activated protein kinase (MAPK) family, consisting of p38, c-Jun N-terminal kinase (JNK), and extracellular-signal-regulated kinase (ERK), belongs to the noncanonical pathway, through which myofibroblasts are activated and lead to the overproduction of extracellular matrix (ECM) proteins [11,12]. Aberrant deposition of the ECM is due to an imbalance between the activities of matrix-degrading proteinases and their inhibitors, which apparently plays an important role in fibrosis [13]. Matrix metalloproteinases (MMPs) belong to a family characterized by zinc-containing endopeptidases that are capable of degrading various ECM proteins, while tissue inhibitors of metalloproteinases (TIMPs) are specific endogenous inhibitors of MMPs. Several studies have reported that TGF-β1 regulates the expression of MMPs and TIMPs, resulting in ECM remodeling in several fibrotic disorders, including systemic sclerosis, idiopathic pulmonary fibrosis, and myelofibrosis [14,15,16]. Alterations in the levels of MMPs, particularly MMP-2 and MMP-9, and TIMPs have been correlated with GO [17,18]. However, the molecular pathways of which are still under investigation.

Connective tissue growth factor (CTGF) is known to participate in ECM remodeling in both physiological and pathological processes [19,20]. Our prior study confirmed the effects of CTGF on the induction of myofibroblast transdifferentiation and ECM production in Graves’ orbital fibroblasts, determined by increased levels of fibrotic markers, including alpha-smooth muscle actin (α-SMA) and fibronectin [6]. In addition, these activities could be enhanced by TGF-β1. In this study, we further investigated the role of the MAPK pathway, including the p38, JNK, and ERK signaling cascades in human Graves’ orbital fibroblasts. The influence of which on the expression of CTGF, fibrotic markers, MMPs, and TIMPs were also examined. Our findings indicated that the p38 and JNK signaling pathways could mediate TGF-β1-induced myofibroblast transdifferentiation and subsequent ECM remodeling in human Graves’ orbital fibroblasts. Moreover, inhibiting p38 or JNK would prevent these fibrogenic processes in GO. 

## 2. Results

### 2.1. TGF-β1 Upregulated p38 and JNK Phosphorylation in Human Graves’ Orbital Fibroblasts

The MAPK pathway transduces signals from TGF-β1 in a number of fibrotic diseases. In the present study, we accessed the contribution of this pathway in TGF-β1-induced fibrogenesis in GO. After treating the orbital fibroblasts from GO patients with TGF-β1 (5 ng/mL), the phosphorylation levels of p38 and JNK, but not ERK, were significantly increased at 3, 6, and 9 h in a time-dependent manner compared with those of the control without TGF-β1 treatment (Figure 1). The results suggested that TGF-β1 specifically activated p38 and JNK in the MAPK pathway in human Graves’ orbital fibroblasts.

### 2.2. p38 and JNK Inhibitors Suppressed TGF-β1-Enhanced Fibrogenesis in GO

CTGF has been proved to mediate TGF-β1-induced fibrogenesis in GO, resulting in the upregulation of α-SMA and fibronectin [6]. In the present study, we measured the expression of CTGF, α-SMA, and fibronectin to evaluate the fibrogenic activities in human Graves’ orbital fibroblasts. After treating the orbital fibroblasts from GO patients with TGF-β1 (5 ng/mL), CTGF, α-SMA, and fibronectin were overexpressed (Figure 2). In order to address the role of the MAPK pathway in the regulation of TGF-β1-mediated fibrogenesis in GO, the specific inhibitors for p38 and JNK were used. After incubating the Graves’ orbital fibroblasts with the p38 inhibitor SB202190 (20 μM) and the JNK inhibitor SP600125 (20 μM), respectively, for 1 h, followed by TGF-β1 (5 ng/mL) treatment for another 9 h, the phosphorylation levels of p38 and JNK were significantly reduced (Appendix A). The inhibition of p38 by SB202190 did not alter TGF-β1-induced JNK phosphorylation. Likewise, the inhibition of JNK by SP600125 did not disturb TGF-β1-induced p38 activation. The results indicated that the TGF-β1-mediated p38 and JNK pathways were independent in the Graves’ orbital fibroblasts. When treating the orbital fibroblasts from GO patients with one of the MAPK inhibitors for 1 h, followed by TGF-β1 (5 ng/mL) for another 24 h, the expression levels of CTGF, α-SMA, and fibronectin were significantly reduced in orbital fibroblasts preincubated with the p38 inhibitor SB202190 (20 μM) and the JNK inhibitor SP600125 (20 μM), respectively, but not in those preincubated with the ERK inhibitor PD98059 (20 μM) (Figure 2). The results suggested that the fibrogenic processes in GO may be prevented by p38 and JNK inhibitors. 

### 2.3. TGF-β1 Affected ECM Metabolism through p38 and JNK Mediators in GO 

The accumulation of ECM proteins is associated with the remodeling of orbital tissues in GO, while MMPs and TIMPs regulate ECM metabolism. Altered levels of MMP-2, MMP-9, and TIMPs have been observed in serum and orbital fibroblasts from GO patients [17,18]. In the present study, we analyzed the protein expression of MMP-2, MMP-9, TIMP-1, and TIMP-3 in human Graves’ orbital fibroblasts. After treatment with TGF-β1 (5 ng/mL), the levels of MMP-2, MMP-9, TIMP-1, and TIMP-3 were increased (Figure 3A,B). When treating with the p38 inhibitor SB202190 (20 μM) and the JNK inhibitor SP600125 (20 μM), respectively, for 1 h, followed by TGF-β1 (5 ng/mL) for another 24 h, the overexpression of TIMP-1 and TIMP-3, but not MMP-2 or MMP-9, was significantly abolished (Figure 3A,B). On the other hand, the activities of MMP-2/-9 were significantly inhibited by TGF-β1 (5 ng/mL) treatment for 24 h, which could be recovered by pretreatment with either the p38 inhibitor SB202190 (20 μM) or the JNK inhibitor SP600125 (20 μM) (Figure 3C). 

## 3. Discussion

The pathological changes of GO primarily involve orbital extraocular muscles and adipose tissues that expand within the unyielding confines of the bony orbit, consequently leading to proptosis, increased orbital pressure, and limited ocular motility [21]. Evidence has shown that TGF-β1 induces Graves’ orbital fibroblasts to differentiate into myofibroblasts, which dominates the pathogenic processes of tissue remodeling and fibrosis [6,7]. This study confirmed that both the p38 and JNK pathways transduced signals from TGF‑β1 to promote myofibroblast transdifferentiation in GO, determined by the increased expression of CTGF, α-SMA, and fibronectin. In consequence, the levels of MMPs and TIMPs were altered in the orbital fibroblasts that affected ECM remodeling. 

The MAPK family belongs to the noncanonical (non-Smad-based) pathway of TGF‑β1 signaling. Normal activation of the MAPK pathway is physiologically essential for stress response and cell survival, while excessive activation of this pathway accelerates inflammation and leads to pathologies [22]. The TGF‑β1-dependent MAPK pathway has been implicated in stimulating myofibroblast transdifferentiation from fibroblasts and subsequent accumulation of the ECM in several fibrotic pathologies. Cui et al. illustrated that p38 and ERK mediated TGF-β1-induced overproduction of the ECM in keloid fibroblasts [23]. Yu et al. proposed that TGF-β1 activated hepatic stellate cells via the p38 and ERK pathways in liver fibrosis [24]. Tian et al. reported the correlation of the TGF-β1-dependent p38 and ERK pathways with nickel oxide nanoparticle-induced pulmonary fibrosis [25]. Additionally, the contribution of the p38 and JNK pathways for TGF-β to activate renal fibrosis has been addressed [26,27]. Therefore, we believed that both the p38 and JNK pathways participated in the pathogenesis of tissue fibrosis in GO.

Although TGF-β1-mediated fibrosis via the noncanonical and canonical signaling pathways has been described in the literature, the cross-talk between these pathways is complex and believed to depend on the cell type and context [28,29]. Additionally, these pathways have not been well-illustrated in orbital fibroblasts before. In addition to the noncanonical pathway, we observed an increase in Smad2/3 phosphorylation in the orbital fibroblasts from GO patients after TGF-β1 treatment (Appendix A). It has been reported that ERK activation may either increase or decrease Smad signaling, while p38 and JNK usually potentiate TGF-β/Smad-induced responses [30]. How Smad-independent TGF-β signaling interacts with the Smad pathway in Graves’ orbital fibroblasts remains to be investigated.

Our findings agreed with previous reports that the subfamilies of the MAPK pathway had respective physiological actions. Briefly, the JNK and p38 pathways were stimulated by stress-related effectors or cytokines to cause inflammatory responses, autophagy, or apoptosis, while the ERK pathway was stimulated by mitogens or growth factors, resulting in cell cycle progression, cell proliferation, and differentiation [31]. Additionally, this study was in accordance with our previous research that CTGF could be a downstream mediator for TGF-β1-induced myofibroblast transdifferentiation and ECM production in Graves’ orbital fibroblasts [6]. This study further showed that p38 and JNK inhibitors could eliminate the effects of TGF-β1 on not only the expression of CTGF, α-SMA, and fibronectin but also ECM metabolism in Graves’ orbital fibroblasts. Notably, TGF-β1-mediated fibrogenesis in the Graves’ orbital fibroblasts is independently transduced through p38 and JNK signaling. It has been reported that the MAPK pathway is involved in the regulation of activating protein-1 (AP-1), a family of pleiotropic transcription factors comprised of Fos, Jun, and activating transcription factor (ATF), that contributes to the control of the fibrotic process [32]. JNK is the only MAPK that can phosphorylate c-Jun, a critical component of the AP-1 complex, while p38 can activate many substrates, including ATF2. The activation of AP-1 is responsible for the TGF-β1-induced α-SMA and fibronectin expression in lung fibrosis [33]. Our present study showed that either p38 inhibitors or JNK inhibitors could completely abolish the TGF-β1-induced expression of α-SMA and fibronectin. We speculate that both p38 and JNK are essential for TGF-β1-mediated fibrosis and that the suppression of either the p38 or JNK pathway may provide a therapeutic potential to correct abnormal orbital tissue remodeling and fibrosis in GO. 

Dysregulation of ECM composition and/or structure is associated with various pathological fibrotic diseases [13,34]. It is especially important that TGF-β contributes to ECM homeostasis by influencing the synthesis and turnover of ECM components through the regulation of matrix-degrading proteolytic enzymes (i.e., MMPs) and their proteinase inhibitors (i.e., TIMPs) [14,35,36]. TIMPs maintain the balance between the deposition and degradation of the ECM in physiological as well as pathological processes [14]. Increased hepatic MMP-2 levels have been associated with liver fibrosis [37]. With regard to GO, MMP-2 and/or MMP-9 and their specific inhibitors have been identified in Graves’ orbital fibroblasts to alter ECM metabolism and subsequent tissue remodeling and fibrosis [17,18]. In the present study, TGF-β1 stimulated the expression of TIMP-1, TIMP-3, MMP-2, and MMP-9 in the Graves’ orbital fibroblasts, which was, however, regulated by p38 and JNK signaling selectively for TIMP-1 and TIMP-3 production. Although TGF-β1 upregulated MMP-2 and MMP-9, it inhibited the activities of MMP-2/-9 in the Graves’ orbital fibroblasts, which could be recovered by inhibiting either p38 or JNK. MMP-2 has been reported to be upregulated by TGF-β1 through Smad2 signaling and Smad3 signaling in fibroblasts and in mesangial as well as endothelial cells, respectively [38,39]. Consistent with our results, overexpression of TIMP-1 and TIMP-3 genes in fibroblasts treated with TGF-β1 was demonstrated, which involved a complex interplay between Smad3, p38, and ERK1/2 signaling [14,40]. Whether TGF-β1 induces the expression of MMP-2 and MMP-9 through the Smad pathway in Graves’ orbital fibroblasts is worth further study. Taken together, TGF-β1-mediated overexpression of TIMP-1 and TIMP-3 as well as inhibition of MMP-2/-9 activities may contribute to abnormal tissue remodeling in GO. Inhibition of p38 or JNK could ameliorate TGF-β1-induced imbalance between ECM deposition and degradation in Graves’ orbital fibroblasts.

Orbital fibroblasts are believed to be the major effector cells in the GO pathogenesis. They are regulated by several inflammatory mediators, including TGF-β, which activates several pathways that contribute to various pathological changes in GO [41]. Thy-1-positive (CD90) orbital fibroblasts have low adipocyte differentiation potential. Rather, they differentiate into myofibroblasts on TGF-β stimulation [7]. Valyasevi et al. reported that TGF-β reduced the expression of thyroid-stimulating hormone receptor (TSHR) in the Graves’ orbital fibroblasts without affecting adipogenesis [42]. Ko et al. revealed that TGF-β upregulated a profibrotic effector, namely sphingosine-1-phosphate (S1P), in the Graves’ orbital fibroblasts [43]. TGF-β affects orbital fibroblast proliferation as well, although there are contrasting results from different studies. Heufelder et al. observed a significant increase in the proliferation of orbital fibroblasts, in particular from patients with GO, after TGF-β stimulation [44]. van Steensel et al. reported that TGF-β had no effect on the proliferation of Graves’ orbital fibroblast but induced hyaluronan synthase gene expression and hyaluronan production [8].

In conclusion, our study provided evidence that p38 and JNK, but not ERK, contributed to the signal transduction of TGF-β1 to activate myofibroblast transdifferentiation and ECM production, through increased expression of TIMP-1 and TIMP-3 and concurrent reduction in MMP-2/-9 activities in human Graves’ orbital fibroblasts. Consequently, blocking the p38 or JNK pathway may be a potential therapeutic target for the prevention or treatment of abnormal tissue remodeling and fibrosis in GO.

## 4. Materials and Methods

### 4.1. Tissue Acquisition and Cell Culture

The surgical specimens of four subjects with GO (GO1–GO4) were recruited during decompression surgery. The disease was in the inactive stage in all the subjects who had achieved stable euthyroidism for at least 6 months prior to surgery by using methimazole. All the subjects were precluded from radiotherapy and systemic corticosteroids for at least 1 month before surgery. The primary cultures of orbital fibroblasts were collected from the surgical orbital tissues in an aseptic technique. Briefly, the orbital tissues were minced and digested with a sterile phosphate-buffered saline (PBS) containing collagenase (130 U/mL) and dispase (1 mg/mL) in an incubator filled with an atmosphere of 5% CO_2_ at 37 °C [6,45]. After 24 h, the digested orbital tissues were collected and resuspended in the cultured medium (DMEM) containing 10% fetal bovine serum (FBS), penicillin G (100 U/mL), and streptomycin sulfate (100 μg/mL). The orbital fibroblasts were grown in the cultured medium and recruited between the 3rd and 5th passages whereby the cell cultures at the same passage number were selected for the same set of experiments.

### 4.2. Chemicals and Antibodies 

The recombinant protein for human TGF-β1 (#P01137) and mouse monoclonal antibodies against human TIMP-1 (#MAB970), TIMP-3 (#MAB973), MMP-2 (#MAB9022), and MMP-9 (#MAB911) were acquired from R&D Systems, Inc (MN, USA). The rabbit polyclonal antibodies against CTGF (#ab6992), fibronectin (#ab2413), and α-SMA (#ab5694) were acquired from Abcam Inc. (Cambridge, UK). The mouse monoclonal antibodies against p38 (#ab31828), p38 phosphorylation (p-p38, #ab4822), JNK (#ab208035), JNK phosphorylation (p-p38, #ab76572), ERK (#ab184699), and ERK phosphorylation (p-p38, #ab201015) were all acquired from Abcam Inc. (Cambridge, UK). The secondary antibodies against rabbit (#A5795) and mouse (#A9044), as well as GAPDH (#G5262) and β-actin (#A5441), were acquired from Sigma-Aldrich (St. Louis, MO, USA) [6]. 

### 4.3. Western Blot Analysis 

The protein was extracted from cell lysates with lysis buffer containing a protease inhibitor cocktail (Bioman, Taipei, Taiwan) as our previous study [45]. An aliquot of 60–100 μg proteins was separated on 10% SDS-PAGE and blotted onto the PVDF membrane (Amersham–Pharmacia Biotech Inc., Buckinghamshire, UK). After blocking with 5% skim milk, the PVDF membrane was incubated with the primary antibody, followed by horseradish peroxidase (HRP)-conjugated antirabbit or antimouse IgG antibody. The protein expression signals were detected by an enhanced chemiluminescence detection kit (Amersham–Pharmacia Biotech Inc., Buckinghamshire, UK) according to the manufacturer’s instruction and were quantified by ImageScanner III with LabScan 6.0 software (GE Healthcare BioSciences Corp., Piscataway, NJ, USA). 

### 4.4. MMP-2 and MMP-9 Enzyme Activity Assay 

The MMP-2/-9 enzyme activities in the culture medium were measured from InnoZyme™ Gelatinase Activity Assay kit (Merck KGaA, Darmstadt, Germany) by a fluorogenic method according to the product instruction [46]. Briefly, the culture medium (100 μL) was diluted in activation buffer and incubated for 3 h with thiopeptide substrate specific for type IV collagenases (i.e., MMP-2 and MMP-9). The released fluorescence of the cleaved substrate of MMP-2/-9 (ex: 320 nm; em: 405 nm) was monitored. Triplicate tests were performed for each reaction in a 96-well plate, and the relative fluorescence unit ratios to the control set were plotted.

### 4.5. Statistical Analysis 

The SPSS software and Microsoft Excel 2019 statistical package were used for statistical analysis. The results from the one-way ANOVA followed by an L.S.D. test and the Student’s *t* test were obtained from three independent experiments, respectively. Data were presented as means ± S.D., and *p* < 0.05 was considered statistically significant.

## Figures and Tables

**Figure 1 ijms-22-02952-f001:**
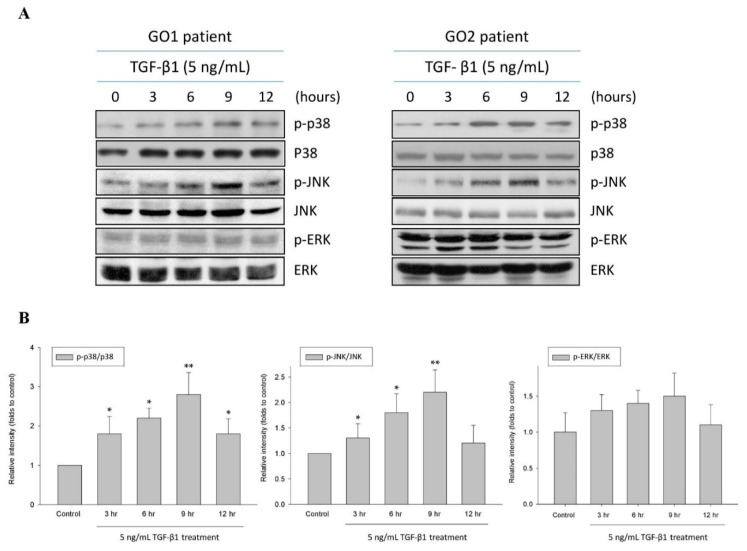
Upregulation of p38 and JNK phosphorylation by TGF-β1 in the primary cultured orbital fibroblasts from patients with Graves’ ophthalmopathy (GO). (**A**) After treating the orbital fibroblasts from GO patients with TGF-β1 (5 ng/mL), the phosphorylation levels of p38, JNK, and ERK were observed for 3, 6, 9, and 12 h by Western blots; (**B**) The expression ratios of p-p38 to p38, p-JNK to JNK, and p-ERK to ERK from the control without TGF-β1 treatment were defined as 1.0. Then, the other relative intensity (folds) was presented. By three independent Western blot experiments, data were averaged together from the same patient strain. The means of the different patient (GO1–GO4, *n* = 4) strains were averaged. The representative histogram was constructed based on the mean values of protein expression levels in the primary cultured orbital fibroblasts from the four GO patients. Data were presented as means ± S.D. of the results from three independent experiments. * *p* < 0.05 vs. control without TGF-β1 treatment; ** *p* < 0.01 vs. control without TGF-β1 treatment.

**Figure 2 ijms-22-02952-f002:**
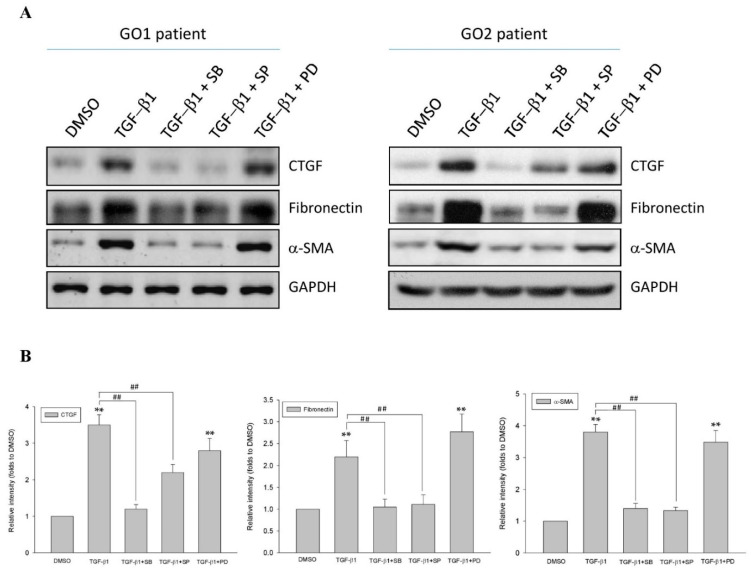
Abolishment of TGF-β1-enhanced fibrotic markers by p38 and JNK inhibitors, but not ERK inhibitors, in the primary cultured orbital fibroblasts from patients with Graves’ ophthalmopathy (GO). (**A**) Orbital fibroblasts from GO patients were preincubated with the p38 inhibitor SB202190 (20 μM), the JNK inhibitor SP600125 (20 μM), and the ERK inhibitor PD98059 (20 μM), respectively, for 1 h, followed by TGF-β1 (5 ng/mL) treatment for another 24 h. Then, the expression levels of CTGF, fibronectin, and α-SMA were analyzed by Western blots; (**B**) The relative intensities of CTGF, fibronectin, and α-SMA expression normalized to each GAPDH control and DMSO control without TGF-β1 treatment were defined as 1.0. Then, the other relative intensity (folds) was presented. By three independent Western blot experiments, together data from the same patient strain were averaged. Then, the means of different patient (GO1–GO4, *n* = 4) strains were averaged. The representative histogram was constructed based on the mean values of protein expression levels in the primary cultured orbital fibroblasts from the four GO patients. Data were presented as means ± S.D. of the results from three independent experiments. ** *p* < 0.01 vs. control without TGF-β1 treatment; ## *p* < 0.01 vs. TGF-β1 treatment.

**Figure 3 ijms-22-02952-f003:**
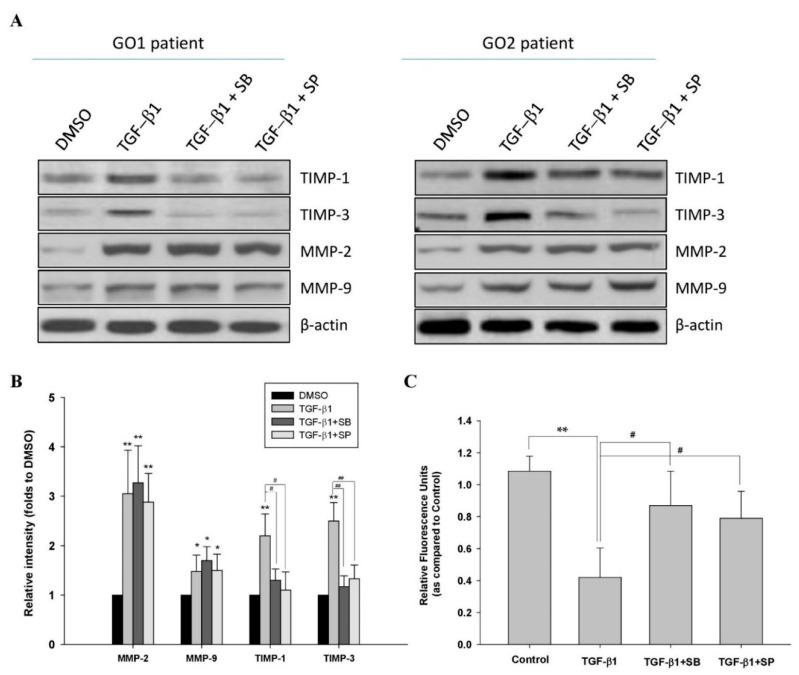
Abolishment of TGF-β1-mediated matrix remodeling by p38 and JNK inhibitors in the primary cultured orbital fibroblasts from patients with Graves’ ophthalmopathy (GO). (**A**) Orbital fibroblasts from GO patients were preincubated with the p38 inhibitor SB202190 (20 μM) and the JNK inhibitor SP600125 (20 μM), respectively, for 1 h, followed by TGF-β1 (5 ng/mL) treatment for another 24 h. Then, the expression levels of TIMP-1, TIMP-3, MMP-2, and MMP-9 were analyzed by Western blots; (**B**) The relative intensities of TIMP-1, TIMP-3, MMP-2, and MMP-9 expression normalized to each GAPDH control and DMSO control without TGF-β1 treatment were defined as 1.0. Then, the other relative intensity (folds) was presented. By three independent Western blot experiments, together data from the same patient strain were averaged. Then, the means of different patient (GO1–GO4, *n* = 4) strains were averaged; (**C**) The enzyme activities of MMP-2/-9 from the cultured medium were determined. The representative histogram was plotted based on the mean values of relative fluorescence units from the primary cultured orbital fibroblasts from the four GO patients. Data were presented as means ± S.D. of the results from three independent experiments. * *p* < 0.05 and ** *p* < 0.01 vs. control without TGF-β1 treatment; # *p* < 0.05 and ## *p* < 0.01 vs. TGF-β1 treatment.

## Data Availability

The data that support the findings of this study are available from the corresponding author upon reasonable request.

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
