# Peer review of "JNK and p38 Inhibitors Prevent Transforming Growth Factor-β1-Induced Myofibroblast Transdifferentiation in Human Graves’ Orbital Fibroblasts"

_ijms, 2021, doi:10.3390/ijms22062952_

Round 1
Reviewer 1 Report
The authors have identified the mechanism through which TGFβ induced myofibroblast transdifferentiation in human orbital fibroblasts from patients with Graves’ ophthalmopathy in vitro. The findings are clearly presented in a series of Western blot experiments performed in GO fibroblasts from 4 donors with the use of pathway specific inhibitors. These findings may suggest a potential therapeutic approach for treatment of fibrosis in GO.
Major concerns.
In this manuscript, the authors state that TGFβ promotes various fibrotic diseases by stimulating either canonical (Smad-based) or non-canonical (non-Smad-based) signaling pathways and focus on the latter studying the non-canonical MAPK kinase pathways, in particular, of p38, JNK, and ERK. Indeed, these pathways have been found Smad-independent but only when there is a rapid (in a course of minutes) phosphorylation of this kinases, that can also be followed by sustained Smad-dependent activity. In the experiments presented in this manuscript, all responses appear at least 3 hours, reaching its peak at 9 hours after stimulation with TGFβ, which indicates there they depend on an intermediate protein upregulation. Therefore, it seems appropriate to check the “usual suspects”, i.e. Smad expression. In my opinion, this would strengthen the study and verify the validity of the statement about the non-canonical TGFβ pathway. Moreover, to support this claim, earlier time points to detect rapid phosphorylation should be included.
Figure 2 shows that phosphokinase inhibitors inhibit their respective kinases, which is what they are designed for and would technically inhibit their phosphorylation regardless of the stimulant. In my opinion, this figure would be better placed in the Supplemental Materials.
Lastly, the authors have demonstrated the importance of p38 and JNK pathways in upregulation of fibrotic markers utilizing their respective inhibitors. It may be important to show pharmacology of the two inhibitors combined: would there be no effect? An additive or synergistic effect? A change in potency? This can help determining whether the two pathways are independent or signal downstream of a common kinase that was not detected in this study.
Minor points.
The fine in graph figures representing relative intensities on the blots is too tiny to see.
Line 99 states that the cells were incubated with the inhibitors for 1 hour, followed by TGFβ treatment for another 9 hours. Could the authors please specify whether the inhibitors were present during the 9-hour treatment with TGFβ?
Tissue Acquisition and Cell Culture section: Please specify what cell culture medium was used to grow orbital fibroblasts.
Author Response
Major concerns.
- Point 1: In this manuscript, the authors state that TGFβ promotes various fibrotic diseases by stimulating either canonical (Smad-based) or non-canonical (non-Smad-based) signaling pathways and focus on the latter studying the non-canonical MAPK kinase pathways, in particular, of p38, JNK, and ERK. Indeed, these pathways have been found Smad-independent but only when there is a rapid (in a course of minutes) phosphorylation of this kinases, that can also be followed by sustained Smad-dependent activity. In the experiments presented in this manuscript, all responses appear at least 3 hours, reaching its peak at 9 hours after stimulation with TGFβ, which indicates there they depend on an intermediate protein upregulation. Therefore, it seems appropriate to check the “usual suspects”, i.e. Smad expression. In my opinion, this would strengthen the study and verify the validity of the statement about the non-canonical TGFβ pathway. Moreover, to support this claim, earlier time points to detect rapid phosphorylation should be included.
Response 1: We appreciate the reviewer's comments. Although cross talk between the Smad-based and non-Smad-based pathways in TGF-β signaling has been well-described, the network is complex and is believed to depend on the cell type and context [1,2]. In orbital fibroblasts, these pathways have not been illustrated in the literature. Our current findings confirmed the involvement of p38 and JNK but not ERK in TGF-β1-induced myofibroblast transdifferentiation in the orbital fibroblasts from GO patients. Indeed, we also observed an increase in the level of Smad2/3 phosphorylation in the orbital fibroblasts from GO patients after TGF-β1 treatment (Supplementary Materials Figure S2). Besides, after TGF-β1 treatment for 1 hour, the phosphorylation levels of p38 and JNK were increased respectively. It has been reported that ERK activation may either increase or decrease Smad signaling, while p38 and JNK usually potentiate TGF-β/Smad-induced responses [3]. How Smad-independent TGF-β signaling interacts with the Smad pathway in Graves’ orbital fibroblasts remains to be investigated in our future work. We have added the above statement in the 3rd paragraph of the Discussion section (line 207–216).
References
- Luo, K. Signaling cross talk between TGF-beta/Smad and other signaling pathways. Cold Spring Harb Perspect Biol. 2017, 9, a022137.
- Derynck, R.; Zhang, Y.E. Smad-dependent and Smad-independent pathways in TGF-beta family signaling. Nature 2003, 425, 577–
- Biernacka, A.; Dobaczewski, M.; Frangogiannis, N.G. TGF-β signaling in fibrosis. Growth Factors 2011, 29, 196–
- Point 2: Figure 2 shows that phosphokinase inhibitors inhibit their respective kinases, which is what they are designed for and would technically inhibit their phosphorylation regardless of the stimulant. In my opinion, this figure would be better placed in the Supplemental Materials.
Response 2: Thanks for the reviewer’s suggestion. In the revised manuscript, this figure is presented as “Figure S1” in the Supplementary Materials. Meanwhile, the remaining figures are numbered as Figure 1, Figure 2, and Figure 3 accordingly. Please refer to the Supplementary Materials Figure S1.
- Point 3: Lastly, the authors have demonstrated the importance of p38 and JNK pathways in upregulation of fibrotic markers utilizing their respective inhibitors. It may be important to show pharmacology of the two inhibitors combined: would there be no effect? An additive or synergistic effect? A change in potency? This can help determining whether the two pathways are independent or signal downstream of a common kinase that was not detected in this study.
Response 3: We appreciate the reviewer's comments. Our Supplementary Materials Figure S1 showed that after pre-treating the primary cultured orbital fibroblasts with the p38 inhibitor, TGF-β1-induced JNK phosphorylation was not altered. Likewise, TGF-β1-induced p38 phosphorylation was not disturbed after pre-treatment with the JNK inhibitor. The result suggested that TGF-β1-mediated p38 and JNK pathways were independent in the Graves’ orbital fibroblasts. On the other hand, the MAPK pathways are involved in the regulation of activating protein-1 (AP-1), a family of pleiotropic transcription factors comprised of Fos, Jun, and activating transcription factor (ATF), that contributes to the control of cell proliferation and survival, as well as fibrotic process [1]. JNK is the only MAPK that could phosphorylate c-Jun, a critical component of the AP-1 complex, while p38 could activate many substrates including ATF2. It has been reported that the activation of AP-1 is responsible for the TGF-β1-induced α-SMA and fibronectin expression during lung fibrosis [2]. Our present study showed that either p38 inhibitors or JNK inhibitors could completely abolish the TGF-β1-induced expression of α-SMA and fibronectin. We speculate that both p38 and JNK are essential for TGF-β1-mediated fibrosis, and that it is not absolutely necessary to combine the p38 and JNK inhibitors to address the additive or synergistic effects. We have revised the Results section “p38 and JNK Inhibitors Suppressed TGF-β1-Enhanced Fibrogenesis in GO” (line 124–133) and added the above statement in the 4th paragraph of the Discussion section (line 226–237).
Reference
- Frangogiannis, N. Transforming growth factor-β in tissue fibrosis. J Exp Med. 2020, 217, e20190103.
- Hu, Y.; Peng, J.; Feng, D.; Chu, L.; Li, X.; Jin, Z.; Lin, Z.; Zeng, Q. Role of extracellular signal-regulated kinase, p38 kinase, and activator protein-1 in transforming growth factor-beta1-induced alpha smooth muscle actin expression in human fetal lung fibroblasts in vitro. Lung 2006, 184, 33–
Minor points.
- Minor Point 1: The fine in graph figures representing relative intensities on the blots is too tiny to see.
Response 1: We have enhanced the resolution of all the figures (Figures 1–3). In addition, we provide the version with high resolution in the zip archive.
- Minor Point 2: Line 99 states that the cells were incubated with the inhibitors for 1 hour, followed by TGFβ treatment for another 9 hours. Could the authors please specify whether the inhibitors were present during the 9-hour treatment with TGFβ?
Response 2: In order to block the appropriate pathway effectively, each inhibitor was added 1 hour prior to the addition of TGF-β1 in the primary cultures of orbital fibroblasts, and was presented during the 9-hour treatment with TGF-β1.
- Minor Point 3: Tissue Acquisition and Cell Culture section: Please specify what cell culture medium was used to grow orbital fibroblasts.
Response 3: The primary cultures of the orbital fibroblasts were cultured in DMEM containing 10% fetal bovine serum (FBS), penicillin G (100 U/mL), and streptomycin sulfate (100 μg/mL). We have made a revision in the Materials and Methods section “Tissue Acquisition and Cell Culture” (line 302–304).
Reviewer 2 Report
I have the pleasure to review the manuscript entitled "JNK and p38 Inhibitors Prevent Transforming Growth Factor-β1-Induced Myofibroblast Transdifferentiation in Human Graves' Orbital Fibroblasts" from Hou and coauthors. This study investigated the role of the mitogen-activated protein kinase (MAPK) pathway in TGF-β1-induced myofibroblast transdifferentiation in human Graves’ orbital fibroblasts. The presented data is interesting and speculative but the research raises serious concerns about methods and sampling. First, the small sample of patients recruited for the study.
Second, the time curve is suddenly abrupt at 12 h. Please, explain your choice. I would recommend expanding the experimentation to 24-72 h to see the time course of kinases expression.
Third, no explanation in the Discussion part on the TGF-b1 in other fibroblasts GO models. For example, according to van Steensel et al. TGF-β has no effect on orbital fibroblast proliferation. I recommend adding efforts on literature search to present more evidence of the authors' approach.
Fourth, the most applicable methods are flow cytometry and real-time PCR in cultivated cells to measure TGF, JNK, ERK expression in cells. Authors are solely relied on Western blots and restricted their work.
Another option to chemical inhibitors is the siRNA approach to inhibit selected pathways.
I recommend using transcriptional assays to confirm the role of MMP-2/-9 proteinases in ECM remodeling.
Author Response
Response to Reviewer 2 Comments
- Point 1: First, the small sample of patients recruited for the study.
Response 1: Thanks for the reviewer’s comment. In this study, we used the primary cultures of orbital fibroblasts from four patients (N=4) with inactive GO who achieved stable euthyroidism for at least 6 months. Several researches utilized the primary cultures of orbital fibroblasts from GO patients for their studies in recent years. They recruited specimens from various numbers of individuals ranging from 3 to 6 [1–7]. Therefore, orbital fibroblasts from 4 GO patients are considered adequate in this study.
Reference
- Wei, Y.H.; Liao, S.L.; Wang, C.C.; Wang, S.H.; Tang, W.C.; Yang, C.H. Simvastatin Inhibits CYR61 Expression in Orbital Fibroblasts in Graves' Ophthalmopathy through the Regulation of FoxO3a Signaling. Mediators Inflamm. 2021, 2021, 8888913. (Orbital fibroblasts from 4 GO patients)
- Kim, J.Y.; Park, S.; Lee, H.J.; Lew, H.; Kim, G.J. Functionally enhanced placenta-derived mesenchymal stem cells inhibit adipogenesis in orbital fibroblasts with Graves' ophthalmopathy. Stem Cell Res Ther. 2020, 11, 469. (Orbital fibroblasts from 3 GO patients)
- Kim, B.R.; Kim, J.; Lee, J.E.; Lee, E.J.; Yoon, J.S. Therapeutic Effect of Guggulsterone in Primary Cultured Orbital Fibroblasts Obtained From Patients with Graves' Orbitopathy. Invest Ophthalmol Vis Sci. 2020, 61, 39. (Orbital fibroblasts from 3 GO patients)
- Lee, J.S.; Kim, J.; Lee, E.J.; Yoon, J.S. Therapeutic Effect of Curcumin, a Plant Polyphenol Extracted From Curcuma longae, in Fibroblasts From Patients With Graves' Orbitopathy. Invest Ophthalmol Vis Sci. 2019, 60, 4129– (Orbital fibroblasts from 3 GO patients)
- Shahida, B.; Johnson, P.S.; Jain, R.; Brorson, H.; Åsman, P.; Lantz, M.; Planck, T. Simvastatin downregulates adipogenesis in 3T3-L1 preadipocytes and orbital fibroblasts from Graves' ophthalmopathy patients. Endocr Connect. 2019, 8, 1230– (Orbital fibroblasts from 4 GO patients)
- Rotondo Dottore, G.; Ionni, I.; Menconi, F.; Casini, G.; Sellari-Franceschini, S.; Nardi, M.; Vitti, P.; Marcocci, C.; Marinò, M. Antioxidant effects of β-carotene, but not of retinol and vitamin E, in orbital fibroblasts from patients with Graves' orbitopathy (GO). J Endocrinol Invest. 2018, 41, 815– (Orbital fibroblasts from 6 GO patients)
- Plöhn, S.; Edelmann, B.; Japtok, L.; He, X.; Hose, M.; Hansen, W.; Schuchman, E.H.; Eckstein, A.; Berchner-Pfannschmidt, U. CD40 Enhances Sphingolipids in Orbital Fibroblasts: Potential Role of Sphingosine-1-Phosphate in Inflammatory T-Cell Migration in Graves' Orbitopathy. Invest Ophthalmol Vis Sci. 2018, 59, 5391-5397. (Orbital fibroblasts from 6 GO patients)
- Point 2: Second, the time curve is suddenly abrupt at 12 h. Please, explain your choice. I would recommend expanding the experimentation to 24-72 h to see the time course of kinases expression.
Response 2: We appreciate the reviewer's comments. Indeed, we did observe the phosphorylation levels of p38, JNK, and ERK in the orbital fibroblasts with TGF-β1 treatment for 24 hours, which turned out to be the same as that of the control without TGF-β1 treatment at 24 hours. In our study, p38 and JNK were phosphorylated immediately at 3 hours after TGF-β1 stimulation, and the levels increased with a peak at 9 hours, followed by a decline at 12 hours. Therefore, we chose the time-point (9 hours) to examine the effects of TGF-β1 on the p38 and JNK signaling pathways.
- Point 3: Third, no explanation in the Discussion part on the TGF-β1 in other fibroblasts GO models. For example, according to van Steensel et al. TGF-β has no effect on orbital fibroblast proliferation. I recommend adding efforts on literature search to present more evidence of the authors' approach.
Response 3: We appreciate the reviewer's suggestion. Orbital fibroblasts have been reported to be the major effector cells in the GO pathogenesis. They are regulated by several inflammatory mediators including TGF-β [1]. Accumulating evidence including the results in our present study demonstrates that TGF-β signaling could activate several pathways that contribute to fibrosis and tissue remodeling in GO. Thy-1-positive (CD90) orbital fibroblasts have low adipocyte differentiation potential. Rather, they can differentiate into myofibroblasts on stimulation with TGF-β [2]. Valyasevi et al. reported that TGF-β reduced the thyroid stimulating hormone receptor (TSHR) expression in the Graves’ orbital fibroblasts without affecting adipogenesis [3]. Ko et al. revealed that TGF-β upregulated sphingosine-1-phosphate (S1P), a profibrotic effector, in the Graves’ orbital fibroblasts [4]. On the other hand, Heufelder et al. observed a significant increase in the proliferation of orbital fibroblasts, in particular from patients with GO, after TGF-β stimulation [5]. However, van Steensel L et al. reported that TGF-β had no effect on the proliferation of Graves’ orbital fibroblast but induced hyaluronan synthase gene expression and hyaluronan production [6]. We have added the above statement in the 6th paragraph of Discussion section (line 271–284).
Reference
- Dik, W.A.; Virakul, S.; van Steensel, L. Current perspectives on the role of orbital fibroblasts in the pathogenesis of Graves' ophthalmopathy. Exp Eye Res. 2016, 142, 83–
- Koumas L, Smith T.J, Feldon S, Blumberg N, Phipps R.P. Thy-1 expression in human fibroblast subsets defines myofibroblastic or lipofibroblastic phenotypes. Am J Pathol, 2003, 163, 1291–1300.
- Valyasevi, R.W.; Jyonouchi, S.C.; Dutton, C.M.; Munsakul, N.; Bahn, R.S. Effect of tumor necrosis factor-alpha, interferon-gamma, and transforming growth factor-beta on adipogenesis and expression of thyrotropin receptor in human orbital preadipocyte fibroblasts. J Clin Endocrinol Metab. 2001, 86, 903–
- Ko, J.; Chae, M.K.; Lee, J.H.; Lee, E.J.; Yoon, J.S. Sphingosine-1-Phosphate Mediates Fibrosis in Orbital Fibroblasts in Graves' Orbitopathy. Invest Ophthalmol Vis Sci. 2017, 58, 2544–
- Heufelder, A.E.; Bahn, R.S. Modulation of Graves' orbital fibroblast proliferation by cytokines and glucocorticoid receptor agonists. Invest Ophthalmol Vis Sci. 1994, 35, 120–
- van Steensel, L.; Paridaens, D.; Schrijver, B.; Dingjan, G.M.; van Daele, P.L.; van Hagen, P.M.; van den Bosch, W.A.; Drexhage, H.A.; Hooijkaas, H.; Dik, W.A. Imatinib mesylate and AMN107 inhibit PDGF-signaling in orbital fibroblasts: a potential treatment for Graves' ophthalmopathy. Invest Ophthalmol Vis Sci. 2009, 50, 3091–
- Point 4: Fourth, the most applicable methods are flow cytometry and real-time PCR in cultivated cells to measure TGF, JNK, ERK expression in cells. Authors are solely relied on Western blots and restricted their work. Another option to chemical inhibitors is the siRNA approach to inhibit selected pathways.
Response 4: We appreciate the reviewer's suggestion. We have demonstrated the gene expression of fibrotic proteins including CTGF, fibronectin and α-SMA in the primary cultures of orbital fibroblasts by a SYBR Green-based real time quantitative PCR (RT-QPCR) in our previous study [1,2]. The results were consistent with our present study that upregulation of the fibrotic proteins analyzed by Western blots was presented in the Graves’ orbital fibroblasts. As to the MAPK pathways, the JNK or p38 activities can also be measured by a commercial ELISA kit from Invitrogen Inc (Catalog Number: 85-86197 and 85-86022, respectively). The MAPK activity assay as well as the siRNA approach to address specifically the role of JNK and p38 will be performed in our further study.
Reference
- Tsai, C.C.; Wu, S.B.; Kau, H.C.; Wei, Y.H. Essential role of connective tissue growth factor (CTGF) in transforming growth factor-β1 (TGF-β1)-induced myofibroblast transdifferentiation from Graves' orbital fibroblasts. Sci Rep. 2018, 8, 7276.
- Tsai, C.C.; Wu, S.B.; Chang, P.C.; Wei, Y.H. Alteration of connective tissue growth factor (CTGF) expression in orbital fibroblasts from patients with Graves’ ophthalmopathy. PLoS One 2015, 10, e0143514.
- Point 5: I recommend using transcriptional assays to confirm the role of MMP-2/-9 proteinases in ECM remodeling.
Response 5: We appreciate the reviewer's suggestion. We have examined the protein expression of MMP-2 and MMP-9 in the primary cultures of orbital fibroblasts after TGF-β1 treatment. The results indicated that TGF-β1 could induce the MMP-2 and MMP-9 expression, which was not disturbed by pre-treatment with either p38 or JNK inhibitors (refer to the revised figures 3A and 3B). Our findings were consistent with the clinical report of the elevated serum concentrations of MMP-2, MMP-9, and TIMP-1 in GO patients [1]. Although upregulation of MMP-2 and MMP-9 by TGF-β1 was observed in the Graves’ orbital fibroblasts, the activities of MMP-2/-9 were significantly inhibited, which could be recovered by pre-treatment with either p38 or JNK inhibitors (refer to figures 3C). With regard to the canonical (Smad-based) pathways, MMP-2 has been reported to be upregulated by TGF-β1 through Smad2 signaling and Smad3 signaling in fibroblasts and in mesangial as well as endothelial cells, respectively [2,3]. Whether TGF-β1 induces the expression of MMP-2 and MMP-9 via the canonical (Smad-based) pathways in Graves’ orbital fibroblasts is worth further study. Besides, TGF-β1 has been shown to induce the TIMP-1 and TIMP-3 gene expression in fibroblasts [4]. Recently, it has been reported that there is a complex interplay between Smad3, p38, and ERK1/2 signaling, through which TGF-β1 induced the TIMP-1 and TIMP-3 expression [5]. In our present study, TGF-β1-mediated overexpression of TIMP-1 and TIMP-3 as well as inhibition of MMP-2/-9 activities may contribute to abnormal tissue remodeling in GO. Most importantly, p38 and JNK inhibitors could ameliorate TGF-β1-induced imbalance between ECM deposition and degradation in Graves’ orbital fibroblasts. We have revised the Results section “TGF-β1 Affected ECM Metabolism through p38 and JNK Mediators in GO” (line 157–165) and added the above statement in the 5th paragraph of the Discussion section (line 249–269).
Reference
- Kapelko–Słowik, K.; Słowik, M.; Szaliński, M.; Dybko, J.; Wołowiec, D.; Prajs, I.; Bohdanowicz–Pawlak, A.; Biernat, M.; Urbaniak–Kujda, D. Elevated serum concentrations of metalloproteinases (MMP-2, MMP-9) and their inhibitors (TIMP-1, TIMP-2) in patients with Graves' orbitopathy. Adv Clin Exp Med. 2018, 27, 99–
- Piek, E.; Ju, W.J.; Heyer, J.; Escalante-Alcalde, D.; Stewart, C.L.; Weinstein, M.; Deng, C.; Kucherlapati, R.; Bottinger, E.P.; Roberts, A.B. Functional characterization of transforming growth factor beta signaling in Smad2- and Smad3-deficient fibroblasts. J Biol Chem. 2001, 276, 19945-19953.
- Kellenberger, T.; Krag, S.; Danielsen, C.C.; Wang, X.F.; Nyengaard, J.R.; Pedersen, L.; Yang, C.; Gao, S.; Wogensen, L. Differential effects of Smad3 targeting in a murine model of chronic kidney disease. Physiol Rep. 2013, 1, e00181.
- Edwards, D.R.; Leco, K.J.; Beaudry, P.P.; Atadja, P.W.; Veillette, C.; Riabowol, K.T. Differential effects of transforming growth factor-β1 on the expression of matrix metalloproteinases and tissue inhibitors of metalloproteinases in young and old human fibroblasts. Exp Gerontol. 1996, 31, 207–
- Leivonen, S.K.; Lazaridis, K.; Decock, J.; Chantry, A.; Edwards, D.R.; Kähäri, V.M. TGF-β-elicited induction of tissue inhibitor of metalloproteinases (TIMP)-3 expression in fibroblasts involves complex interplay between Smad3, p38α, and ERK1/2. PLoS One 2013, 8, e57474.
Round 2
Reviewer 1 Report
I thank the authors for responded to the comments. I believe the manuscript has been sufficiently improved and ready for publication.
Reviewer 2 Report
Authors addressed all the necessary issues. I recommend this manuscript to publication.